# Phosphorylation of PSD-95 at serine 73 in dCA1 is required for extinction of contextual fear

**Magdalena Ziółkowska**[1☯], **Malgorzata Borczyk**[1,2☯], **Anna Cały**[1], **Kamil F. Tomaszewski**[1], **Agata Nowacka**[1], **Maria Nalberczak-Skóra**[1], **Małgorzata Alicja Śliwińska**[1,3], **Kacper Łukasiewicz**[1,4], **Edyta Skonieczna**[1], **Tomasz Wójtowicz**[5], **Jakub Wlodarczyk**[5], **Tytus Bernaś**[3,6], **Ahmad Salamian**[1], **Kasia Radwanska**[1] *

1 Laboratory of Molecular Basis of Behavior, the Nencki Institute of Experimental Biology of Polish Academy of Sciences, Warsaw, Poland, 2 Department Molecular Neuropharmacology, Maj Institute of Pharmacology of Polish Academy of Sciences, Krakow, Poland, 3 Laboratory of Imaging Tissue Structure and Function, the Nencki Institute of Experimental Biology of Polish Academy of Sciences, Warsaw, Poland, 4 Psychiatry Clinic, Medical University of Bialystok, Białystok, Poland, 5 Laboratory of Cell Biophysics, the Nencki Institute of Experimental Biology of Polish Academy of Sciences, Warsaw, Poland, 6 Department of Anatomy and Neurology, VCU School of Medicine, Richmond, Virginia, United States of America

☯ These authors contributed equally to this work.
* k.radwanska@nencki.edu.pl

**Data Availability Statement:** All relevant data is available OSF (https://osf.io/cgfa9/) DOI 10.17605/OSF.IO/CGFA9.

## Abstract

The updating of contextual memories is essential for survival in a changing environment. Accumulating data indicate that the dorsal CA1 area (dCA1) contributes to this process. However, the cellular and molecular mechanisms of contextual fear memory updating remain poorly understood. Postsynaptic density protein 95 (PSD-95) regulates the structure and function of glutamatergic synapses. Here, using dCA1-targeted genetic manipulations in vivo, combined with ex vivo 3D electron microscopy and electrophysiology, we identify a novel, synaptic mechanism that is induced during attenuation of contextual fear memories and involves phosphorylation of PSD-95 at Serine 73 in dCA1. Our data provide the proof that PSD-95–dependent synaptic plasticity in dCA1 is required for updating of contextual fear memory.

## Introduction

The ability to form, store, and update memories is essential for animal survival. In mammals, the formation, recall, and updating of memories involve the hippocampus [1–3]. In particular, formation of memories strengthens the Schaffer collateral-to-dorsal CA1 area (dCA1) synapses through N-methyl-D-aspartate receptor (NMDAR)-dependent forms of synaptic plasticity [4–6] linked with growth and addition of new dendritic spines (harbouring glutamatergic synapses) [7–10]. Although some studies also found long-term depression of synaptic transmission during hippocampal-dependent tasks [11,12]. Similarly, updating and extinction of memories induces functional, structural, and molecular alterations of dCA1 synapses [13–15]. Accordingly, NMDAR-dependent plasticity of dCA1 synapses is commonly believed to be a primary

**Funding:** This work was supported by a National Science Centre (Poland) (Grant SONATA BIS No. 2015/19/B/NZ4/02996 and grant MAESTRO No. 2020/38/A/NZ4/00483 to KR; Grant PRELUDIUM No. 2016/21/N/NZ4/03304 to MZ; Grant PRELUDIUM No. 2015/19/N/NZ4/03611 to KŁ; Grant PRELUDIUM No. 2019/35/N/NZ4/01910 to KFT; Grant SONATA BIS No. 2017/26/E/NZ4/00637 to JW; Grant SONATA BIS No. 2019/34/E/NZ4/00387 to TW). The funders had no role in study design, data collection and analysis, decision to publish, or preparation of the manuscript.

**Competing interests:** The authors have declared that no competing interests exist.

**Abbreviations:** ACSF, artificial cerebrospinal fluid; CaMKII, calmodulin-dependent kinase II; CFC, contextual fear conditioning; CLEM, correlative light-electron microscopy; dCA1, dorsal CA1; EC, entorhinal cortex; fEPSP, field excitatory postsynaptic potential; NMDAR, N-methyl-D-aspartate receptor; PSD, postsynaptic density; PSD-95, postsynaptic density protein 95; Re, nucleus reuniens; ROI, region of interest; RT, room temperature; S73, Serine 73; SBEM, serial block-face scanning electron microscopy; stLM, stratum lacunosum-moleculare; stOri, stratum oriens; stRad, stratum radiatum; WT, wild-type.

cellular learning mechanism. Surprisingly, the role of dCA1 synaptic plasticity in memory formation has been recently questioned. Local genetic manipulations that impair synaptic function and plasticity specifically in dCA1 affect spatial choice and incorporation of salience information into cognitive representations, rather than formation of cognitive maps and memory engrams [16–20]. On the other hand, the role of dCA1 synaptic plasticity in the updating and extinction of existing hippocampus-dependent memories has not been tested yet. Understanding the molecular and cellular mechanisms that underlie fear extinction memory is crucial to develop new therapeutic approaches to alleviate persistent and unmalleable fear memories.

Postsynaptic density protein 95 (PSD-95) is the major scaffolding protein at glutamatergic synapses [21]. It directly interacts with NMDARs and with AMPARs through an auxiliary protein, stargazin [22,23]. Interaction of PSD-95 with stargazin regulates the synaptic content of AMPARs [23–25]. Accordingly, PSD-95 affects stability and maturation as well as functional and structural plasticity of glutamatergic synapses [26–35]. Synaptic localisation of PSD-95 is controlled by a range of posttranslational modifications with opposing effects on its synaptic retention as well as synaptic function and plasticity [36]. Here, in order to test the role of dCA1 excitatory synapses in extinction of fear memories, we focused on phosphorylation of PSD-95 at Serine 73 (S73). PSD-95(S73) is phosphorylated by the calcium and calmodulin-dependent kinase II (CaMKII) [32,37]. Expression of phosphorylation-deficient PSD-95, with S73 mutated to Alanine [PSD-95(S73A)], blocks the reduction in the NMDAR/PSD-95 interaction during chemical LTP in a manner that is dependent on CaMKII and calpain [38]. Hence, phosphorylation of PSD-95(S73) enables PSD-95 dissociation from the complex with GluN2B, and its trafficking to regulate synaptic growth after stimulation of NMDA receptors, and is necessary for PSD-95 protein down-regulation during NMDAR-dependent long-term depression of synaptic transmission (LTD) [32,39]. Importantly, both authophosphorylation-deficient αCaMKII mutant mice (αCaMKII-T286A) [40] and the loss-of-function PSD-95 mutants lacking the guanylate kinase domain of PSD-95 [26] show impaired extinction of contextual fear [9,41], suggesting that αCaMKII and PSD-95 interact to regulate contextual fear extinction.

The present study tests the role of PSD-95(S73) phosphorylation in the dorsal hippocampus in fear memory extinction by integrated analyses of PSD-95 protein expression and phosphorylation, dCA1-targeted expression of phosphorylation-deficient PSD-95 protein (with S73 mutated to alanine, S73A), as well as examination of dendritic spines morphology with nanoscale resolution enabled by electron microscopy. We show that phosphorylation of PSD-95 (S73) is necessary for contextual fear extinction-induced PSD-95 protein regulation and remodelling of glutamatergic synapses. Moreover, it is not necessary for fear memory formation but required for fear extinction even after extensive fear extinction training. Overall, our data show for the first time that the dCA1 PSD-95(S73) phosphorylation is required for extinction of the contextual fear memory.

## Results

### The contextual fear extinction affects PSD-95 protein levels and morphology of dendritic spines in stOri dCA1

To investigate the role of dCA1 excitatory synapses in contextual fear memory extinction, we trained Thy1-GFP(M) mice (that allow for visualisation of dendritic spines) [42] in contextual fear conditioning (CFC). The animals showed low freezing levels in the novel context before delivery of 5 electric shocks (US), after which the freezing levels increased during the rest of the training session (**Fig** 1A). Twenty-four hours later, one group of mice was killed (5US)

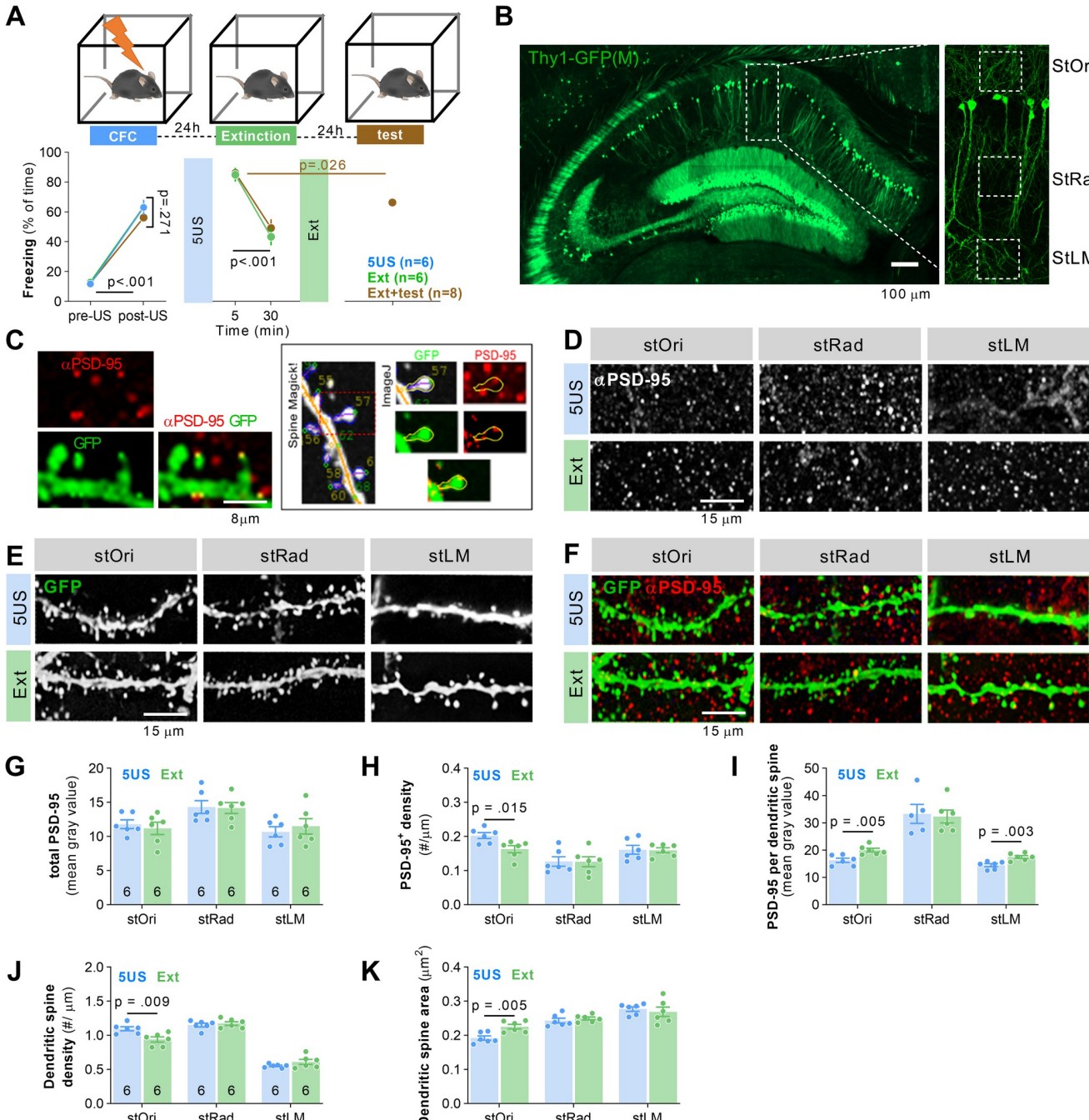

**Fig 1. Extinction of contextual fear memory regulates PSD-95 protein levels and remodelling of dendritic spines in stOri dCA1. (A)** Experimental timeline and freezing levels during training. Mice underwent CFC and were killed 24 hours later (5US, $n = 6$) or after reexposure to the training context without electric shocks (Ext, $n = 6$) (two-way repeated-measures ANOVA, effect of training: $F(1, 10) = 77.86$, $P < 0.0001$). **(B, C)** Dendritic spines and PSD-95 expression were analysed in 3 domains of the dendritic tree of dCA1 pyramidal neurons (stOri, stRad, and stLM) in Thy1-GFP(M) male mice. **(B)** Microphotography of dCA1 and dendritic tree domains. **(C)** High magnification of confocal scans showing colocalization of PSD-95 immunostaining and dendritic spines, and the analysis in SpineMagick! and ImageJ. **(D-F)** Representative confocal images (maximum projections of z-stacks composed of 20 scans) of PSD-95 immunostaining, GFP and their colocalization are shown for 3 domains of dCA1. **(G-I)** Summary of data showing total PSD-95 expression (two-way repeated-measures ANOVA with Tukey's multiple comparisons test (marked on the graphs), effect of training: $F(2, 14) = 1.126$, $P = 0.3521$), density of PSD-95$^+$ puncta (effect of training: $F(2, 13) = 1.30$, $P = 0.305$), and area of PSD-95$^+$ puncta (effect of training: $F(2, 15) = 5.653$, $P = 0.015$). **(J, K)** Summary of data showing dendritic spine density (effect of training: $F(2, 44) = 2.851$, $P = 0.069$; a region effect: $F(1.983, 43.63) = 6.293$, $P = 0.004$; training × region interaction: $F(4, 44) = 5.389$, $P = 0.001$) and average dendritic spine area (two-way repeated-measures ANOVA with Tukey's multiple comparisons test; effect of training: $F(2, 42) = 1.630$, $P = 0.208$; a region effect: $F(2, 42) = 46.49$, $P < 0.001$; training × region interaction: $F(4, 42) = 2.121$, $P = 0.095$). The analyses were conducted in stOri (mouse/dendrite/spine: 5US = 6/25/650; Ext = 6/37/925). For G-I, each dot represents one mouse.

For A and G-K, means ± SEM are shown. The data underlying this figure are available from OSF (https://osf.io/cgfa9/). CFC, contextual fear conditioning; dCA1, dorsal CA1; PSD-95, postsynaptic density protein 95; stLM, stratum lacunosum-moleculare; stOri, stratum oriens; stRad, stratum radiatum.

(mice were randomly assigned to the experimental groups), and the second group was reexposed to the training context for 30 minutes without presentation of US for extinction of contextual fear (Ext). Freezing levels were high at the beginning of the session and decreased within the session, indicating formation of fear extinction memory (t = 3.720, df = 6, $P < 0.001$). Mice were killed immediately after the fear extinction session. Twenty-four hours later, the third group of mice was reexposed to the training context (without US) to test consolidation of fear extinction memory (test). Freezing levels were lower during the test as compared to the beginning of the extinction, indicating that our protocol resulted in efficient fear extinction ($P = 0.026$) (Fig 1A). The mouse brains were sliced, the brain sections immunostained to detect PSD-95 protein using specific antibodies, and imaged with a confocal microscope. The scans were analysed to assess PSD-95 protein levels [total PSD-95 as mean grey value of microphotographs; density of PSD-95–positive puncta (PSD-95+) per 1 μm of a dendrite and mean grey value of PSD-95+ per dendritic spine] (Fig 1C). As dendritic spines change in dCA1 after CFC in a dendrite-specific manner [10], the expression of PSD-95 protein, and its colocalization with dendritic protrusions, were analysed in 3 domains of dCA1: stratum oriens (stOri), stratum radiatum (stRad), and stratum lacunosum-moleculare (stLM) (Fig 1B).

The analysis of the confocal scans revealed that there were no differences between the experimental groups in total PSD-95 levels (analysed as mean grey value of microphotographs) in 3 strata of dCA1 (Fig 1G). However, there were less PSD-95+ puncta after fear extinction, as compared to the 5US group, in stOri, but not in other dCA1 strata (Fig 1H). There was also a significant effect of the training on the mean grey value of PSD-95+ per dendritic spine. In the stOri, the mean grey value of PSD-95+ increased after extinction, as compared to the 5US group (Fig 1I), indicating bidirectional PSD-95 changes (elimination of PSD-95+ puncta and increased content of PSD-95 in the remaining puncta). No difference in PSD-95+ area was observed between the groups in stRad, and an increase of mean grey value of PSD-95+ in stLM.

Next, we checked whether the changes in PSD-95 protein levels were associated with dendritic spine remodelling. In stOri, dendritic spine density decreased after extinction as compared to the 5US mice (Fig 1J). No changes in dendritic spine density were observed in the stRad and stLM. Moreover, the median dendritic spine area was increased in stOri after extinction, compared to the 5US group, resembling the changes of PSD-95 protein levels. No changes in the median dendritic spine area were observed in the stRad and stLM (Fig 1K).

To confirm the fear extinction–induced synaptic changes in stOri, we used serial block-face scanning electron microscopy (SBEM) that allows for reconstruction of dendritic spines and postsynaptic densities (PSDs), representing a postsynaptic part of excitatory synapses, with nanoscale resolution [43]. We determined dendritic spine density using unbiased brick method [44] and reconstructed dendritic spines and PSDs to assess dendritic spine and PSD volume (as a proxy of the accumulated synaptic proteins [45]), as well as PSDs surface area (as a proxy of synaptic strength [46–48]) (Fig 2A–2C). In total, we calculated density of dendritic spines for 5 animals per experimental group (3 tissue bricks per animal) and reconstructed 386 spines from the brains of C57BL/6J mice killed 24 hours after CFC (5US) ($n = 3$), and 447 spines from the mice killed after fear extinction (Ext) ($n = 3$). We observed that dendritic spine density was significantly decreased in the Ext mice as compared to the 5US groups (Fig 2D), while median dendritic spine volume, PSD volume and area were significantly increased after

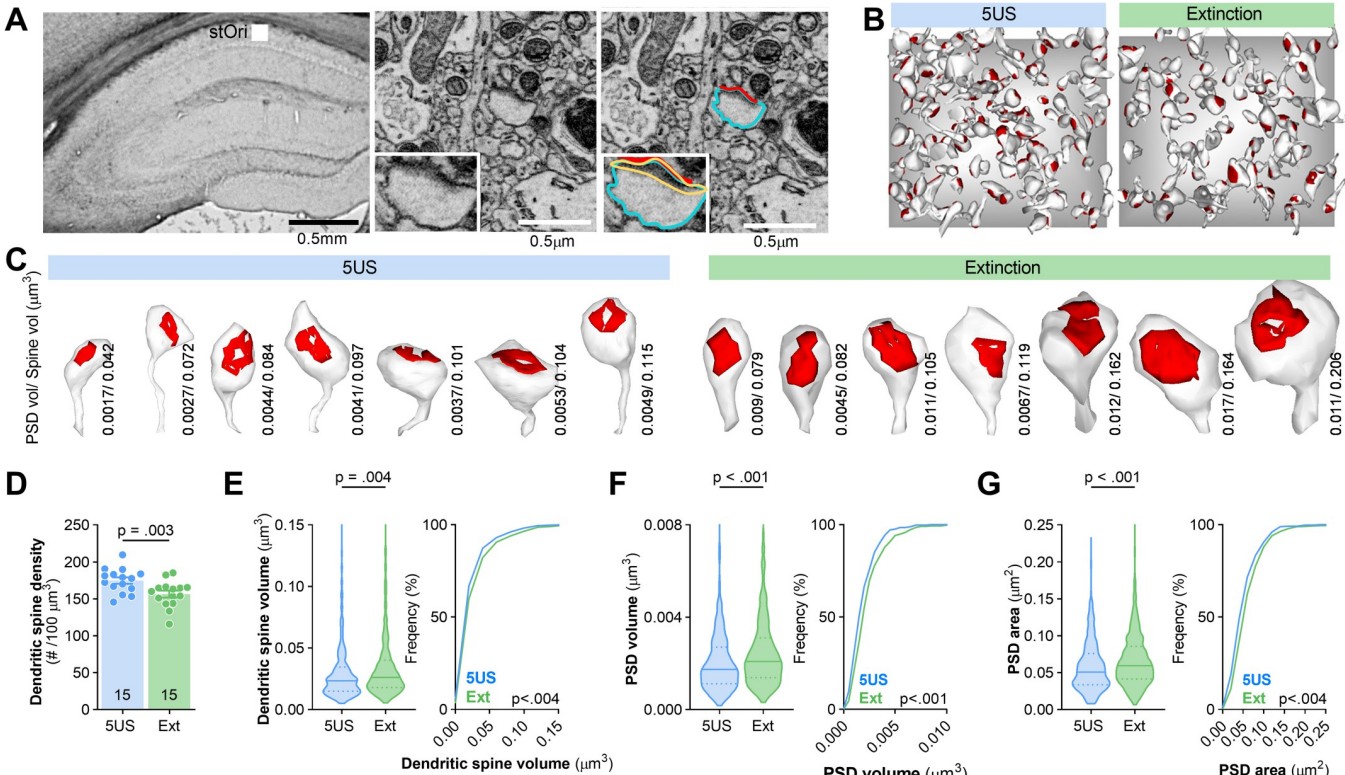

**Fig 2. Extinction of contextual fear memory remodels dendritic spines and PSDs in stOri dCA1. (A)** The principles for SBEM analysis of the ultrastructure of dendritic spines and PSDs. **(Left)** Microphotography of a dorsal hippocampus with the region of interest for analysis and tracing of a dendritic spine and PSD in stOri. **(Middle/right)** A representative trace of a dendritic spine (blue), PSD surface area (red), and volume (yellow). **(B, C)** Exemplary reconstructions of dendritic spines and their PSDs from SBEM scans in stOri. **(B)** Dendritic spines and PSDs were reconstructed and analysed in tissue bricks ($3 \times 3 \times 3$ μm). The grey background squares are x = $3 \times$ y = 3 μm. **(C)** Exemplary reconstructions of dendritic spines and PSDs (red). PSD and dendritic spine volumes are indicated. **(D-G)** Summary of SBEM data showing: **(D)** density of dendritic spines (*t* test, t(28) = 2.94, $P = 0.003$); **(E)** median volume (Mann–Whitney U = 74,533, $P < 0.001$) and distribution of dendritic spine volumes (Kolmogorov–Smirnov D = 0.123, $P = 0.004$); **(F)** median volume (Mann–Whitney U = 70,978, $P < 0.001$) and distribution of PSD volumes (Kolmogorov–Smirnov D = 0.152, $P < 0.001$); **(G)** median PSD surface area (Mann–Whitney U = 70,306, $P < 0.001$) and distribution of values (Kolmogorov–Smirnov D = 0.151, $P < 0.001$). For D, each dot represents one tissue brick. For D, means ± SEM are shown. For E-G (left), medians ± IQR are shown. The data underlying this figure are available from OSF (https://osf.io/cgfa9/). dCA1, dorsal CA1; PSD, postsynaptic density; SBEM, serial block-face scanning electron microscopy; stOri, stratum oriens.

extinction (**Fig** 2E–2G, left). We also observed that the distributions of the values for dendritic spine volume, PSD volume, and area were shifted towards bigger values in Ext group compared with the 5US (**Fig** 2E–2G, right), confirming the dendritic spine changes observed in stOri of Thy1-GFP(M) mice.

In a separate experiment, we found that dendritic spine changes observed in Thy1-GFP mice were transient, as they were not observed 60 minutes after contextual fear extinction session, and they were specific for fear extinction, as we did not find such changes in the animals exposed to a neutral novel context (not associated with US) as compared to 5US group (**S1 Fig**). Overall, our data indicate that contextual fear extinction involves transient remodelling of the stOri neuronal circuit characterised by decreased density of dendritic spines with PSD-95 as well as up-regulation of PSD-95 protein levels, dendritic spine volume, PSD volume, and area in the remaining dendritic spines. No significant synaptic changes of these parameters were found in stRad, and only the increase of mean grey value of PSD-95[+] puncta was observed in stLM.

## Contextual fear extinction induces phosphorylation of PSD-95(S73) in dCA1

Phosphorylation of PSD-95(S73) has been associated with regulation of PSD-95 levels during LTP and LTD [37,39]. To test whether contextual fear extinction induces phosphorylation of PSD-95(S73) in dCA1, we generated an antibody directed against this phosphorylation site (LERGN**S**GLGFS sequence) (**Fig** 3A) [37]. Mice underwent CFC and were killed 24 hours later (5US), or after 15 or 30 minutes of the contextual fear extinction session (Ext15' or Ext30') (**Fig** 3B). The levels of PSD-95, phosphorylated PSD-95(S73) [phospho-PSD-95(S73)] and their colocalization were tested on the brain sections (**Fig** 3C). Total PSD-95, phospho-PSD-95(S73), and their colocalization levels were higher in the Ext15' group, but not Ext30' group, as compared to the 5US animals (**Fig** 3D–3F). Thus, our data indicate that the alteration of PSD-95 protein levels during contextual fear extinction was accompanied by transiently increased phosphorylation of PSD-95(S73). The important limitation of this experiment is the fact that, using phospho-S73 antibody, we cannot exclude that other MAGUKs are detected (due to the similar LERGN**S**GLGFS sequence). However, the role of phospho-PSD-95(S73) in contextual fear extinction is supported by the fact that there is increased colocalization of PSD-95 and phospho-PSD-95(S73) during extinction.

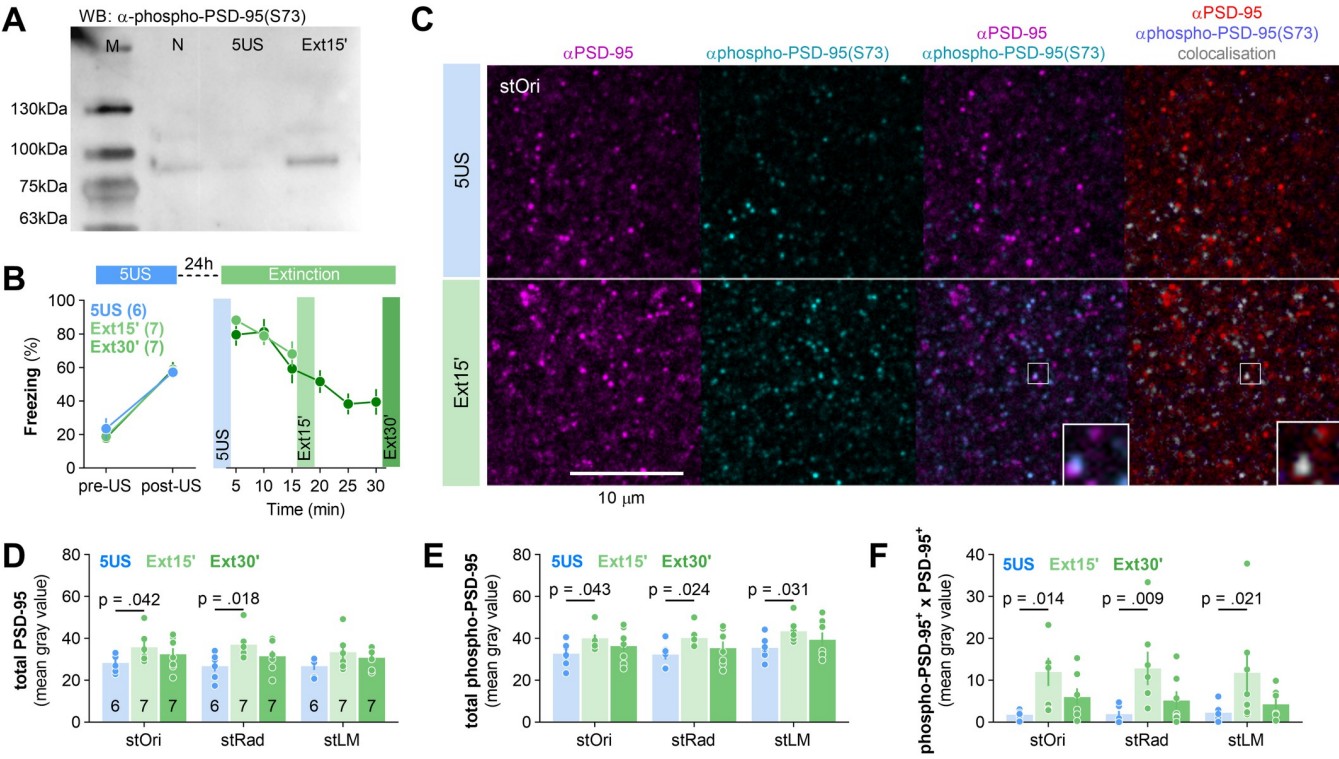

**Fig 3. Contextual fear extinction induces transient phosphorylation of PSD-95(S73) in dCA1. (A)** Western blot stained with phospho-PSD-95(S73)-specific antibody detects in the hippocampus homogenates proteins with approximately 95 kDA molecular weight. M, molecular weight marker; N, naive mouse. **(B)** Experimental timeline and freezing levels during training. Mice underwent CFC and were killed 24 hours later (5US, $n = 6$) or after 15 or 30 minutes of a fear extinction session (Ext15', $n = 7$; Ext30', $n = 7$). **(C)** Representative confocal scans of the brain slices (stOri) immunostained with antibodies specific for PSD-95, phosphorylated PSD-95(S73), and their colocalization. **(D-F)** Quantification of the PSD-95 (two-way ANOVA, effect of training: $F(2, 17) = 2.69$, $P = 0.097$; effect of stratum: $F(1,96, 33,3) = 3.83$, $P = 0.033$), phospho-PSD-95(S73) (two-way ANOVA, effect of training: $F(2, 17) = 2.20$, $P = 0.141$; effect of stratum: $F(1,24, 21,0) = 24.9$, $P < 0.001$) and their colocalization levels (two-way ANOVA, effect of training: $F(2, 17) = 4.08$, $P = 0.036$; effect of stratum: $F(2, 34) = 0.169$, $P = 0.845$). Each dot represents one mouse. Means ± SEM are shown. The data underlying this figure and raw image for A are available from OSF (https://osf.io/cgfa9/).CFC, contextual fear conditioning; dCA1, dorsal CA1; PSD-95, postsynaptic density protein 95; S73, Serine 73; stOri, stratum oriens.

## PSD-95(S73) phosphorylation regulates PSD-95 protein levels during contextual fear extinction

To test whether phosphorylation of PSD-95(S73) regulates PSD-95 protein levels in dCA1 during fear extinction, we used dCA1-targeted expression of phosphorylation-deficient PSD-95 (S73A). We designed and produced adeno-associated viral vectors (AAV1/2) encoding wild-type (WT) PSD-95 protein under *Camk2a* promoter fused with mCherry (AAV1/2:CaM-KII_PSD-95(WT):mCherry) (WT) or PSD-95(S73A) fused with mCherry (AAV1/2:CaM-KII_PSD-95(S73A):mCherry) (S73A) [39] (**S2 Fig**). Mice underwent CFC (**Fig** 4A). The animals in all experimental groups showed increased freezing levels at the end of the training. Half of the mice were killed 24 hours after CFC (5US). The remaining half were killed after the 30-minute contextual fear extinction session (Ext). All animals showed high freezing levels at the beginning of the session, which decreased during the session. No effect of the virus on animal behaviour was found (**Fig** 4A).

For each animal, half of the brain was chosen at random for confocal analysis of the total PSD-95 protein levels, and the other half was processed for SBEM (**Fig** 4B). The AAVs penetrance did not differ between the experimental groups (5US versus Ext) and reached over 80% of the cells in the analysed sections of dCA1 (**Fig** 4C). We observed a significant increase in total PSD-95 protein levels in WT and S73A mice killed before the fear extinction session as compared to the Control group killed at the same time point (**S2A–S2C Fig**). Correlative light and electron microscopy confirmed that the exogenous PSD-95 colocalised with PSDs and weak signal was present in dendrites (**Fig** 3D). Furthermore, overexpression of PSD-95 protein (WT and S73A) resulted in decreased dendritic spines density and increased surface area of PSDs, compared to the Control group. However, total PSD surface area per tissue brick was not changed (**S2D–S2G Fig**).

As in Thy1-GFP mice, the total PSD-95 protein levels were not changed after fear extinction in the WT group, as compared to the WT mice killed before the fear extinction session (**Fig** 4E and 4F). However, PSD-95 levels were up-regulated in all strata after the extinction session in the S73A mice, as compared to the WT Ext animals and the S73A 5US group (**Fig** 4F). Hence, exogenous PSD-95(S73A) protein impaired regulation of PSD-95 levels in dCA1 during contextual fear extinction, indicating that phosphorylation of PSD-95(S73) controls PSD-95 levels during this process.

## Phosphorylation of PSD-95(S73) regulates stOri synapses during fear extinction

To test whether phosphorylation of PSD-95(S73) regulates structural plasticity of excitatory synapses during contextual fear extinction, we used SBEM. We reconstructed dendritic spines and PSDs in the stOri. In total, we reconstructed 159 spines from the brains of the WT mice killed 24 hours after CFC (5US, *n* = 3), and 178 spines from the mice killed after fear extinction (Ext) (*n* = 3). For mice expressing S73A, 183 spines were reconstructed in the 5US group (*n* = 3) and 160 in the Ext (*n* = 3). Figs 4C and 5A show reconstructions of dendritic spines from representative SBEM brick scans for each experimental group.

Dendritic spine density was lower in the WT Ext group, as compared to the WT 5US mice (**Fig** 5B). Furthermore, the median and summary (per volume of tissue) dendritic spine volume, PSD surface area, and PSD volume were higher after the extinction training in the WT group, as compared to the WT 5US mice. These changes were also indicated as shifts in the frequency distributions towards bigger values of all analysed metrics (**Fig** 5D-5F). Overall, the pattern of synaptic changes observed in the WT mice after contextual fear extinction resembled the changes found in C57BL/6J animals (**Fig** 2). In addition, field excitatory postsynaptic

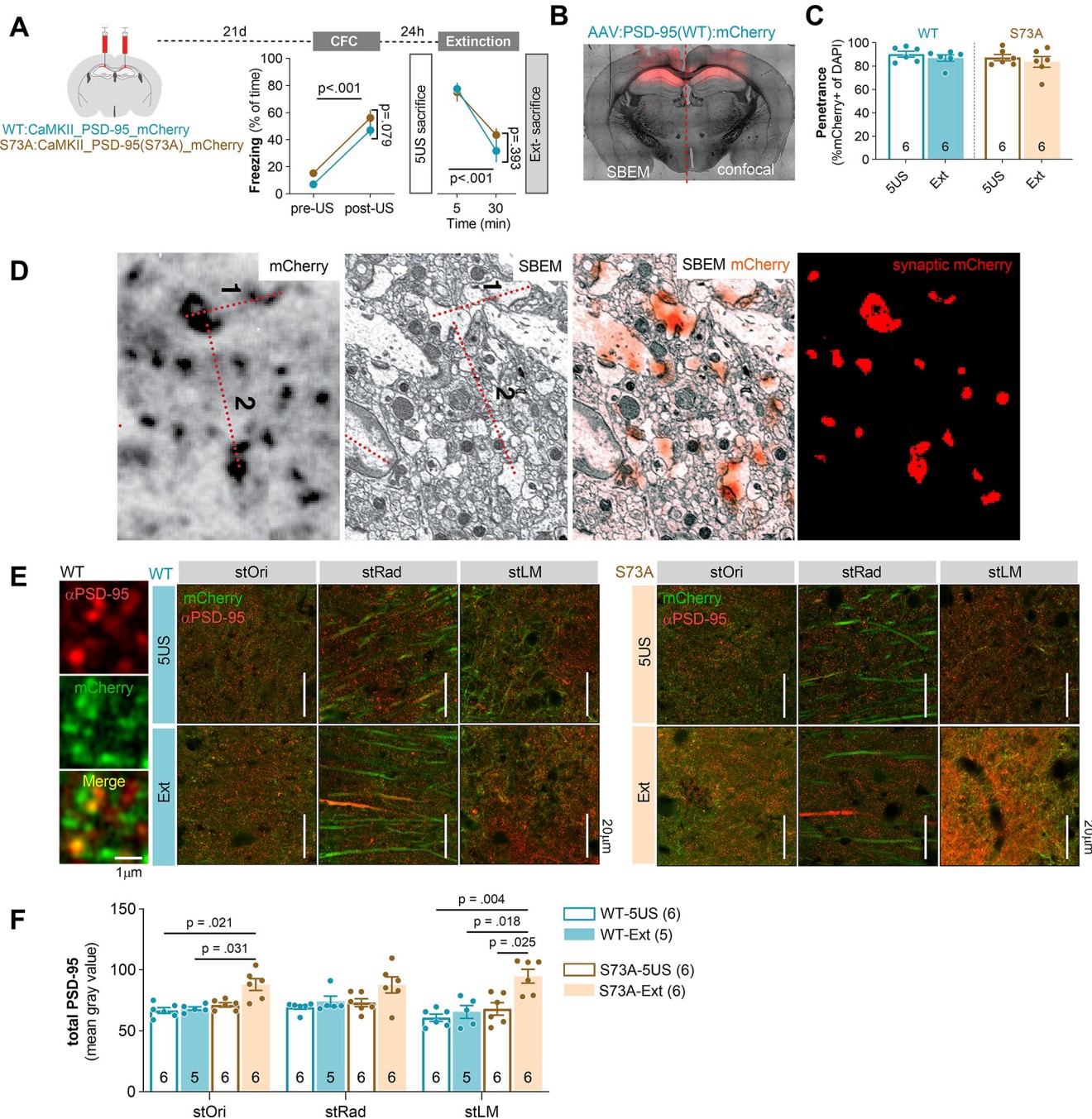

**Fig 4. PSD-95(S73) is phosphorylated during fear extinction and this process is required for regulation of PSD-95 protein levels. (A)** Experimental timeline and freezing during training. C57BL/6J male mice were stereotactically injected in the dCA1 with AAV1/2 encoding PSD-95(WT) (WT, $n$ = 12) or PSD-95(S73A) (S73A, $n$ = 12). Twenty-one days later, they underwent CFC (two-way repeated-measures ANOVA, effect of training: $F_{(1, 30)}$ = 269.4, $P$ < 0.001, effect of virus: $F_{(2, 30)}$ = 2.815, $P$ = 0.076) and were killed 1 day after training (5US) or they were reexposed to the training context without footshock and killed (Ext) (two-way repeated-measures ANOVA, effect of training: $F_{(1, 15)}$ = 65.68, $P$ < 0.001; effect of virus: $F_{(2, 15)}$ = 0.993, $P$ = 0.393). **(B)** Microphotography of a brain with dCA1 PSD-95(WT):mCherry expression with illustration of the brain processing scheme. **(C)** Summary of data showing the viruses penetrance in dCA1 (sections used for confocal and SBEM analysis) (mice: 5US/Ext, WT = 6/5; S73A = 6/6). **(D)** Correlative confocal-electron microscopy analysis showing that exogenous PSD-95(WT) colocalizes with PSDs. Single confocal scan of an exogenous PSD-95(WT) in dCA1, SBEM scan of the same area, superposition of confocal (orange) and SBEM images based on measured distances between large synapses (1 and 2), and thresholded synaptic PSD-95(WT) signal. Measurements: (confocal image) 1: 3.12 μm, 2: 4.97 μm; (SBEM image) 1: 2.98 μm, 2: 4.97 μm. **(E, F)** Analysis of total PSD-95 expression after fear extinction training. **(E)** Representative confocal scans of the PSD-95 immunostaining and **(F)** summary of data showing total PSD-95 levels (tree-way ANOVA with LSD post hoc tests for planned comparisons, effect of training × virus interaction, $F_{(1, 19)}$ = 4.603, $P$ = 0.0451). Means ± SEM are shown. The data underlying this figure are available from OSF (https://osf.io/cgfa9/). CFC, contextual fear conditioning; dCA1, dorsal CA1; PSD-95, postsynaptic density protein 95; S73, Serine 73; SBEM, serial block-face scanning electron microscopy; WT, wild-type.

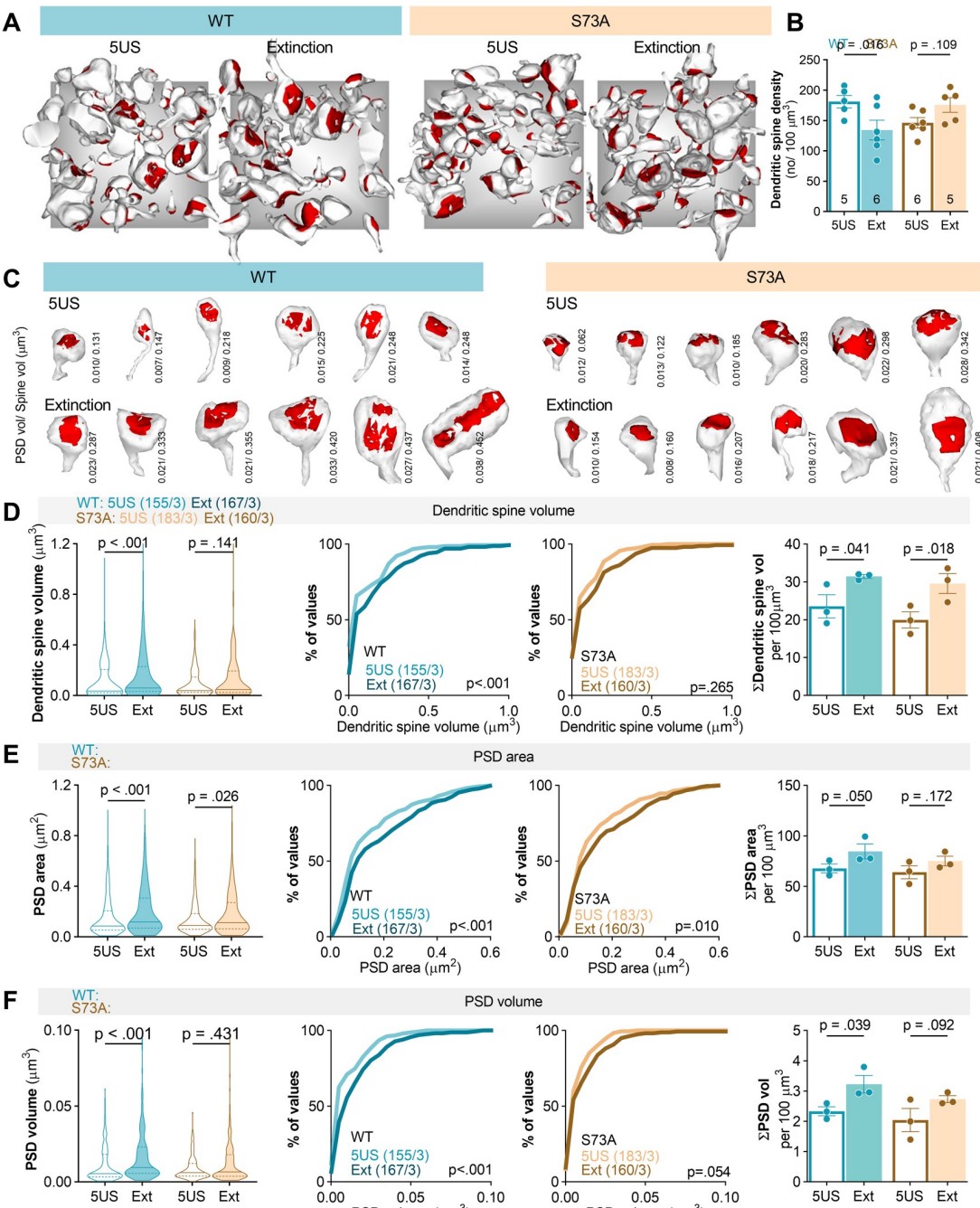

**Fig 5. Phosphorylation of PSD-95(S73) regulates excitatory synapses during fear extinction.** Male mice were stereotactically injected in the dCA1 with AAV1/2 encoding PSD-95(WT) (WT, $n = 12$) or PSD-95(S73A) (S73A, $n = 12$). Twenty-one days later, they underwent CFC and were killed 1 day after training (5US) or they were reexposed to the training context for fear extinction (Ext). **(A)** Exemplary reconstructions of dendritic spines and their PSDs from SBEM scans in stOri tissue bricks ($3 \times 3 \times 3$ μm). The grey background rectangles are x = $3 \times$ y = 3 μm. **(B)** Summary of data showing mean density of dendritic spines (two-way ANOVA with LSD post hoc tests for planned comparisons, effect of training × genotype interaction: F(1, 18) = 9.42; $P = 0.007$). Each dot represents one tissue brick. **(C)** Exemplary reconstructions of dendritic spines and PSDs (red). PSD and dendritic spine volumes are indicated for each dendritic spine. **(D-F)** Summary of data showing: **(D)** median dendritic spine volume (Mann–Whitney test, WT: U = 9,766, $P < 0.001$; S73A: U = 13,217, $P = 0.141$), distributions of dendritic spine volumes (numbers of the analysed dendritic spines/mice are indicated) (Kolmogorov–Smirnov test, WT: D = 0.239, $P < 0.001$; S73A: D = 0.109, $P = 0.265$) and summary dendritic spine volume per tissue brick (two-way ANOVA with LSD post hoc tests for planned comparisons; effect of training, F(1, 8) = 14.6, $P = 0.005$; effect of genotype, F(1, 8) = 1.41, $P = 0.269$); **(E)** median PSD surface area (Mann–Whitney test, WT: U = 9,948, $P < 0.001$; S73A: U = 46,678, $P = 0.024$), distributions of PSD surface areas

(numbers of the analysed dendritic spines/mice are indicated) (Kolmogorov–Smirnov test, WT: D = 0.157, $P < 0.001$; S73A: D = 0.128, $P = 0.010$), and summary PSD surface area per tissue brick (two-way ANOVA with LSD post hoc tests for planned comparisons; effect of training, $F(1, 8) = 5.71$, $P = 0.044$; effect of genotype, $F(1, 8) = 1.31$, $P = 0.285$); **(F)** median PSD volume (Mann–Whitney test, WT: U = 9,462, $P < 0.001$; S73A: U = 13,621, $P = 0.431$), distributions of PSD volumes (numbers of the analysed dendritic spines/mice are indicated) (Kolmogorov–Smirnov test, WT: D = 0.278, $P < 0.001$; S73A: D = 0.145, $P = 0.054$), and summary PSD volume per tissue brick (two-way ANOVA with LSD post hoc tests for planned comparisons; effect of training, $F(1, 8) = 9.56$, $P = 0.015$; effect of genotype, $F(1, 8) = 2.35$, $P = 0.164$). The data underlying this figure are available from OSF (https://osf.io/cgfa9/). CFC, contextual fear conditioning; dCA1, dorsal CA1; PSD, postsynaptic density; PSD-95, postsynaptic density protein 95; S73, Serine 73; SBEM, serial block-face scanning electron microscopy; stOri, stratum oriens; WT, wild-type.

potentials (fEPSPs) were measured in stOri of the acute hippocampal slices of the WT Ext and 5US mice when Shaffer collaterals were stimulated by monotonically increasing stimuli. The input–output curves showed significant increase in the amplitude of fEPSP in the WT mice killed after fear extinction as compared to the WT 5US group (**S3C Fig**). As no changes in fibre volley were observed, our data indicate that contextual fear extinction resulted in global increase in synaptic strength in stOri of WT mice.

On the other hand, S73A mutation impaired fear extinction–induced down-regulation of dendritic spine density (**Fig** 5B). We also found no significant changes of median dendritic spine volumes and PSD volumes (**Fig** 5D and 5F), and only a minor increase in the median PSD surface area in the S73A Ext group as compared to the S73A 5US animals (**Fig** 5E). These impairments were confirmed by the analyses of the distributions of metrics values (**Fig** 5D–5F). We also found that mutation prevented an increase of summary PSD volume and surface area, while the increase of summary dendritic spine volume after fear extinction was preserved (**Fig** 5D–5F, right panels). In addition, we observed no difference in fEPSP and fibre volley between the S73A mice killed before versus after fear extinction session (**S3D Fig**). Altogether, our data indicate that PSD-95(S73) phosphorylation regulates density, size, and strength of the excitatory synapses in stOri during contextual fear extinction.

## PSD-95(S73) phosphorylation in dCA1 is required for extinction of contextual fear

To test whether phosphorylation of PSD-95(S73) is necessary for consolidation of fear extinction memory, we used dCA1-targeted expression of S73A, WT, or control AAV1/2 encoding mCherry under *Camk2a* promoter (Control). Two cohorts of mice with dCA1-targeted expression of the Control virus, WT, or S73A underwent CFC and fear extinction training. The first cohort underwent a short extinction training with one 30-minute extinction session (Ext) and 5-minute test of fear extinction memory (Test) (**Fig** 6A), while the second underwent an extensive fear extinction training with three 30-minute contextual fear extinction sessions on the days 2, 3, 4 (Ext1 to 3), followed by spontaneous fear recovery/remote fear memory test on day 18, and further 3 extinction sessions on the days 18 to 20 (Ext4 to 6). Next, fear generalisation was tested in a context B (CtxB, day 22) (**Fig** 6D). The posttraining analysis showed that the viruses were expressed in dCA1. The control virus was expressed in 85% of the dCA1 cells, WT in 88%, and S73A in 87% (**Fig** 6I and 6J).

The analysis of the short extinction training (data pooled from 2 cohorts) showed that in all experimental groups freezing levels were low at the beginning of the training and increased after 5US delivery (**Fig** 6B). Furthermore, mice in all groups showed high freezing levels at the beginning of the Ext indicating similar levels of contextual fear memory acquisition. However, freezing measured during the Test was significantly decreased, as compared to the beginning of Ext, only in the Control and WT groups, not in the S73A animals (**Fig** 6C).

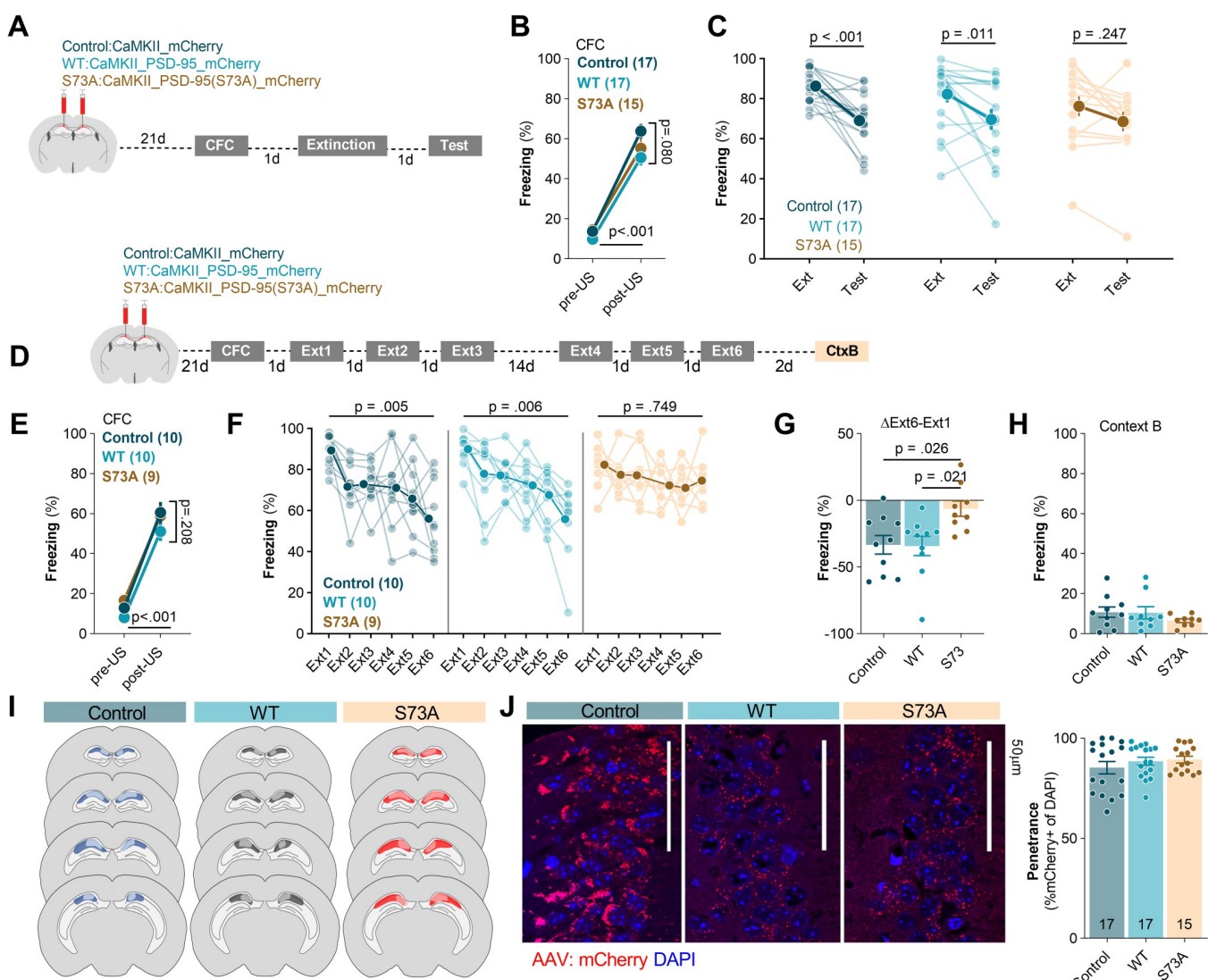

**Fig 6. Phosphorylation of PSD-95(S73) in dCA1 is required for contextual fear extinction. (A)** Experimental timeline of the short fear extinction training. C57BL/6J male mice were stereotactically injected in the dCA1 with AAV1/2 encoding mCherry (Control, $n = 17$), PSD-95(WT) (WT, $n = 17$) or PSD-95 (S73A) (S73A, $n = 15$). Twenty-one days after surgery mice underwent CFC. One day after CFC, they were reexposed to the training context in the absence of foot shock (Extinction). Consolidation of fear extinction memory was tested 1 day later in the same context (Test). **(B, C)** Summary of data showing percentage of freezing during **(B)** CFC, **(C)** extinction and test of the mice with dCA1-targeted expression of Control, WT, or S73A (two-way repeated-measures ANOVA with Šidák's multiple comparisons test, effect of time: $F(1, 46) = 26.13$, $P < 0.001$, genotype: $F(2, 46) = 0.540$, $P = 0.586$; time x genotype: $F(2, 46) = 1.25$, $P = 0.296$). **(D)** Experimental timeline of the extensive fear extinction training. Mice with dCA1-targeted expression of Control ($n = 10$), WT ($n = 10$), or S73A ($n = 9$) underwent CFC, followed by six 30-minute fear extinction sessions (Ext1–6) and one exposure to novel context without footshock (CtxB). **(E-H)** Summary of data showing freezing levels **(E)** during CFC, **(F)** after extensive fear extinction training (two-way repeated-measures ANOVA with Dunnett's multiple comparisons test, effect of time: $F(3.681, 95.70) = 13.01$, $P < 0.001$; genotype: $F(2, 26) = 1.23$, $P = 0.306$; time x genotype: $F(10, 130) = 1.49$, $P = 0.147$), **(G)** the difference in freezing between Ext1 and Ext6 (one-way ANOVA with Tukey's multiple comparisons test, $F(2, 24.94) = 4.98$, $P = 0.016$), and **(H)** during the test in the context B (Brown–Forsythe ANOVA test, $F(2, 17.56) = 0.902$, $P = 0.428$). **(I)** The extent of viral infection. **(J)** Single confocal scans of the stratum pyramidale of dCA1 of the mice expressing Control, WT, and S73A and penetrance of the viruses. Means ± SEM are shown. The data underlying this figure are available from OSF (https://osf.io/cgfa9/). CFC, contextual fear conditioning; dCA1, dorsal CA1; PSD-95, postsynaptic density protein 95; S73, Serine 73; WT, wild-type.

The analysis of freezing levels during the extensive fear extinction training showed high levels of freezing at the end of training and beginning of Ext1 for all experimental groups (**Fig 6E and 6F**). In the Control and WT groups, the freezing levels decreased over consecutive extinction sessions (Ext2 to 6) and were significantly lower as compared to Ext1, indicating

formation of long-term fear extinction memory. We also found no spontaneous fear recovery after 14-day delay (Ext4 versus Ext3; Control, $P = 0.806$; WT, $P = 0.248$). In the S73A group, the extensive contextual fear extinction protocol did not reduce freezing levels measured at the beginning of Ext6 sessions, as compared to Ext1, indicating no fear extinction (**Fig** 6F). Accordingly, we found significantly larger reduction of freezing after fear extinction training (ΔExt6-Ext1) in the controls and WT animals, as compared to the S73A group (**Fig** 6G). The freezing reaction was specific for the training context, as it was very low and similar for all experimental groups in the context B (**Fig** 6H). Thus, our data indicate that expression of the S73A in dCA1 does not affect fear memory formation, recall, or generalisation but prevents contextual fear extinction even after extensive fear extinction training.

## αCaMKII autophosphorylation regulates contextual fear extinction and PSD-95 protein levels during contextual fear extinction

PSD-95(S73) is phosphorylated by αCaMKII [32,37]. To test the role of αCaMKII in fear extinction and PSD-95 protein regulation during fear extinction, autophosphorylation-deficient αCaMKII mutant mice (T286A) [40] and their WT littermates (males and females, in sex-balanced groups) were trained in CFC. They had similar and low levels of freezing in the novel context and freezing increased after 5US delivery (**Fig** 7A). Mice of both genotypes also showed high levels of freezing in the training context on the next day (Ext1), indicating contextual fear memory formation. However, when the mice were reexposed to the training context for fear extinction (Ext2 to 3), the freezing levels of WT mice were significantly lower, as compared to Ext1, while the T286A mutants showed still high freezing. Thus, we confirmed that αCaMKII autophosphorylation is required for contextual fear memory extinction.

Next, a second cohort of WT and T286A mice was trained, and the animals were killed 24 hours after training (5US) or after fear extinction session (Ext). The total levels of PSD-95 were not affected by the fear extinction session in WT mice (**Fig** 7C). However, PSD-95 levels were higher in all strata of dCA1 in the T286A Extinction group, as compared to WT animals killed after extinction, and T286A mutants killed before extinction. Thus, these experiments support the hypothesis that αCaMKII autophosphorylation is required for fear extinction and extinction-induced regulation of PSD-95 levels in dCA1.

## Discussion

We have investigated the role of dCA1 PSD-95(S73) phosphorylation in contextual fear extinction. Our study showed that (1) contextual fear extinction induces transient changes of dCA1 PSD-95 protein levels and dendritic spines in a stratum-specific manner, and these changes are observed mostly in stOri; (2) contextual fear extinction induces phosphorylation of PSD-95(S73) in all dCA1 strata; (3) expression of the exogenous, phosphorylation-deficient PSD-95 (S73A) in dCA1 deregulates PSD-95 protein levels and synaptic plasticity (both structural and functional) induced by extinction of fear memories; (4) dCA1 PSD-95(S73A) impairs contextual fear extinction memory, but not fear memory formation or recall; and (5) phosphorylation-deficient αCaMKII(T286A) impairs contextual fear extinction and regulation of dCA1 PSD-95 protein levels during fear extinction.

We demonstrate that contextual fear extinction transiently increases phospho-PSD-95(S73) levels and induces rapid down-regulation of the synapses with PSD-95 as well as growth of the remaining synapses in stOri. Such synaptic plasticity alludes to the previously reported Hebbian strengthening of activated synapses and heterosynaptic weakening of adjacent synapses [49,50]. To our knowledge, our study is the first demonstration of such synaptic plasticity during attenuation of fear memories. To study the role of this plasticity in fear memory extinction,

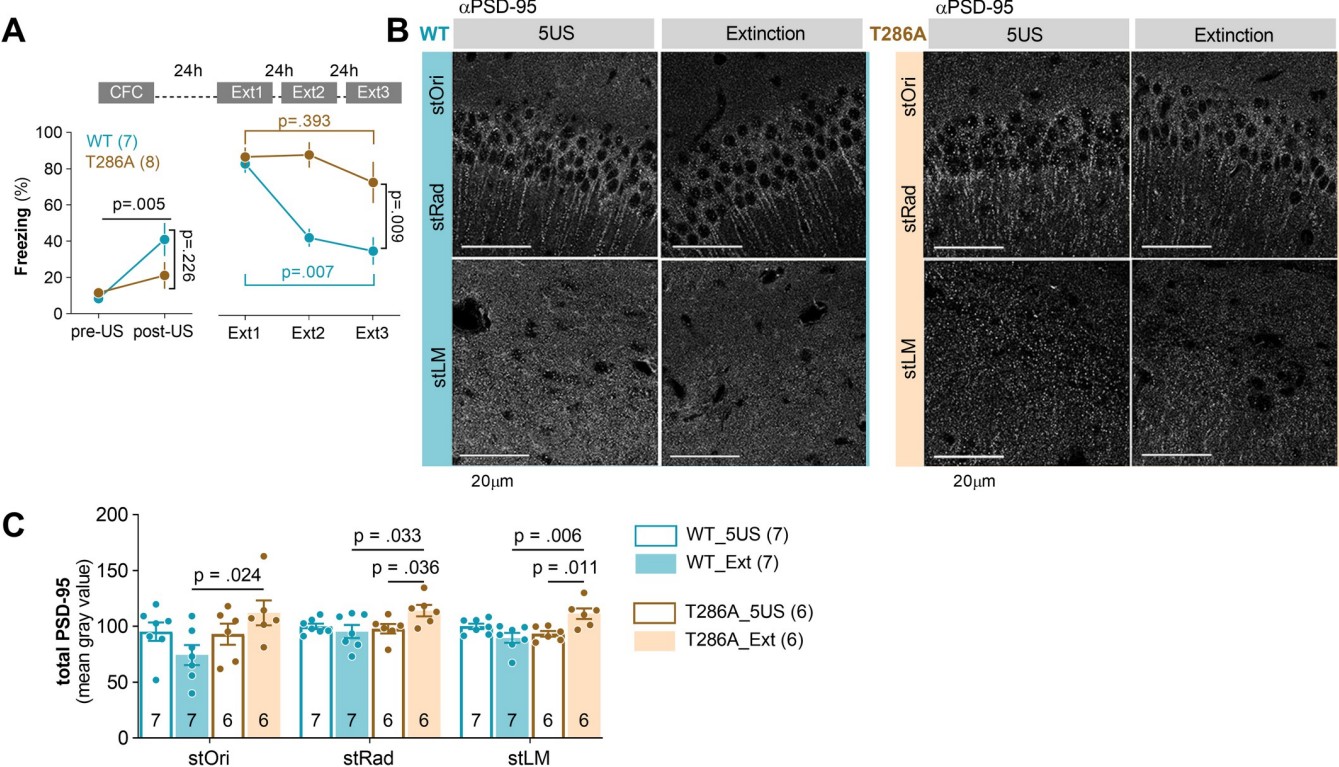

**Fig 7. Autophosphorylation of αCaMKII is required for extinction of contextual fear and regulation of PSD-95 levels during fear extinction training. (A)** Experimental timeline [WT and T286A underwent CTC and 3 fear extinction sessions (Ext1–3)] and percentage of freezing during CFC (two-way repeated-measures ANOVA, effect of time: $F_{(1, 10)} = 13.06$, $P = 0.005$; effect of genotype: $F_{(1, 10)} = 1.66$, $P = 0.226$) and Ext1–3 (WT/T286A = 7/8; sex-balanced groups) (two-way repeated-measures ANOVA with Šídák's multiple comparisons test, effect of training: $F_{(1.430, 18.59)} = 14.96$, $P < 0.001$; effect of genotype: $F_{(1, 13)} = 9.30$, $P = 0.009$). **(B)** Representative confocal scans of the WT and T286A brain slices immunostained to detect PSD-95. **(C)** Quantification of the total PSD-95 protein levels (three-way repeated-measures ANOVA with with Tukey's post hoc test, effect of genotype x training: $F_{(1, 22)} = 15.03$, $P < 0.001$); Mice: 5US/Ext, WT = 7/7; T286A = 6/6. Means ± SEM are shown. The data underlying this figure are available from OSF (https://osf.io/cgfa9/). CFC, contextual fear conditioning; PSD-95, postsynaptic density protein 95; WT, wild-type.

we used dCA1-targeted overexpression of PSD-95(WT) and PSD-95(S73A). Interestingly, although PSD-95(WT) overexpression reduced dendritic spine density and increased median spine and PSD volume before fear extinction, the extinction-induced synaptic processes were preserved in PSD-95(WT) mice and fear extinction resulted in elimination of dendritic spines, as well as global increase of dendritic spine volume, PSD size, and strengthening of synaptic transmission. Hence, our research contradicts the earlier in vitro studies showing that PSD-95 overexpression results not only in synaptic growth, but also increased density of dendritic spines and impaired LTP [28,51]. These differences between our study and in vitro experiments may result from the spatial constraints that are imposed on the neurons in the brain, but not in vitro, as well as the fact that neurons in vivo coexist in complex networks that may induce homeostatic/compensatory mechanisms. Still, taking into account significant changes of dendritic spine morphology induced by PSD-95 overexpression in vivo, it is likely that computational power of such neurons is affected [52]. As we found no effect of PSD-95(WT) overexpression on fear memory formation, recall, extinction, or generalisation, future studies are needed to discover whether any cognitive processes are affected by this presumed limitation of computational properties of the dCA1 neurons. It is also possible that our study further supports the notion that synaptic plasticity in dCA1 is not required for the formation of long-term memory [16,19].

On the other hand, extinction-induced down-regulation of stOri synapses, as well as regulation of PSD-95 protein levels, synaptic growth, and strengthening, are impaired by the expression of phosphorylation-deficient PSD-95(S73A) and αCaMKII-T286A mutation, which prevents phosphorylation of PSD-95(S73). Moreover, the local genetic manipulation prevented contextual fear extinction. These observations indicate that phosphorylation of PSD-95 (S73), and PSD-95-dependent synaptic plasticity, are important steps in the regulation of the dCA1 circuit during fear extinction. Importantly, using ex vivo analyses, we cannot unequivocally indicate whether PSD-95(S73A) prevents extinction-induced elimination of dendritic spines and PSD-95 proteins or changes the balance of the synapses by enhancing synaptogenesis and protein synthesis. We believe, however, that the first scenario is more likely and that this conclusion is supported by several observations. Firstly, PSD-95(S73) phosphorylation allows for dissociation of PSD-95 from the complex with GluN2A, destabilisation and remodelling of PSD [32,37,38], as well as NMDA-induced down-regulation of PSD-95 levels [39]. Secondly, both dCA1 phosphorylation of PSD-95(S73) and protein degradation, but not protein synthesis, are necessary for contextual fear extinction [53,54].

To our knowledge, our experiments are the first to show that phosphorylation of PSD-95 (S73) in dCA1 is required for extinction of contextual fear memories. Strikingly, the contextual fear memory cannot be updated even when the animals with dCA1 PSD-95(S73A) mutation undergo six 30-minute extinction sessions. We also show that dCA1 PSD-95(S73A) does not affect mice activity, long-term fear memory formation and recall, context-independent fear generalisation or fear recovery after 14-day delay, pointing towards engagement of PSD-95 (S73) phosphorylation only during extinction of contextual fear. This conclusion seemingly contradicts the study demonstrating that ligand binding-deficient PSD-95 knock-in mice have enhanced contextual fear memory formation and impaired long-term memory retention [41,55]. However, even though the behavioural phenotype of PSD-95 KI mice was supported by LTP analysis in dCA1 [41,55], it is unknown whether the mouse phenotype relies on the CA1 plasticity as the mutation was global. Furthermore, it is possible that PSD-95 KI and PSD-95(S73A) impact different stages of contextual fear memory. In agreement with our findings, the signalling pathways downstream of NMDAR-PSD-95 complex in the dorsal CA3 and DG regulate contextual fear extinction [56,57]. In particular, translocation of PSD-95 from NMDAR to TrkB, and increased PSD-95-TrkB interactions, promote extinction, while competing NMDAR-PSD-95-nNOS interactions hinder contextual fear extinction [56]. Since PSD-95(S73A) mutation prolongs NMDAR-PSD-95 interactions [37], it may limit interactions of PSD-95 with TrkB and fear extinction. To support this hypothesis, we also show that autophosphorylation of αCaMKII, the key enzyme activated by NMDAR, is required for extinction-induced regulation of PSD-95 levels and fear extinction.

Our data show that the extinction of contextual fear affects PSD-95 protein levels and dendritic spines predominantly in the stOri. This indicates that the extinction-induced synaptic remodelling is strikingly different from the changes observed immediately after contextual fear memory encoding where transient synaptogenesis is observed in the stRad [9]. These observations support the idea that different CA1 inputs are involved in memory formation and extinction. CA3 neurons project to the stRad and stOri regions of CA1 pyramidal neurons; the nucleus reuniens (Re) projects to the stOri and stLM; and the entorhinal cortex (EC) projects to the stLM [58–61]. Thus, the pattern of synaptic changes induced by contextual fear extinction colocalizes with the domains innervated by the Re and EC, suggesting that these inputs are regulated during contextual fear extinction. In agreement with our observations, previous data showed that the EC is activated during and required for contextual fear extinction in animal models [1,62]. Human studies also showed that EC-CA1 projections are activated by cognitive prediction error (that may drive memory extinction), while CA3-CA1 projections are

activated by memory recall without prediction errors [63]. The role of the Re in fear memory encoding, retrieval, extinction, and generalisation has been demonstrated [64–66]. Still, it has to be established whether the plasticity of dCA1 synapses is specific to Re and/or EC projections.

Our findings add up to the previous studies investigating the molecular processes in dCA1 that are specific and required for contextual fear extinction, but not for fear memory consolidation, including regulation of ERK, CB1, and CBEP [67–72]. Interestingly, other processes, such as protein synthesis and c-Fos expression, are necessary for contextual fear consolidation and reconsolidation, but not extinction [53,67,73,74]. Thus, although it is not surprising that distinct molecular cascades and cell circuits contribute to fear memory formation/recall and extinction [67,75], it remains puzzling how synaptic plasticity, without concomitant translation, contributes to contextual fear extinction. This observation points towards the role of protein synthesis-independent short-term plasticity, or protein degradation [54], in contextual fear extinction memory. The role of short-term plasticity in contextual fear extinction is supported by the observations that PSD-95(S73) phosphorylation and synaptic remodelling induced by fear extinction are transient. Similar short plasticity was observed by other groups upon recall of drug-paired memories [76,77]. Still, it has to be clarified in the future studies how short-term dCA1 plasticity can support long-term fear extinction memory.

## Conclusions

Our study demonstrates that extinction of contextual fear memories relies on rapid and transient synaptic plasticity in dCA1 that requires PSD-95(S73) phosphorylation. Thus, our study supports the hypothesis that NMDAR-dependent plasticity in dCA1 is required to detect and resolve contradictory or ambiguous memories when spatial information is involved [17], the comparator view of hippocampal function [78,79] as well as the observations that the hippocampus processes surprising events and prediction errors [63,80–82]. Since new or long-lasting memories may be repeatedly reorganised upon recall [83,84], the molecular and cellular mechanisms involved in extinction of the existing fearful memories provide excellent targets for fear memory impairment therapies. In particular, understanding the mechanisms that underlie contextual fear extinction may be relevant for posttraumatic stress disorder treatment.

## Materials and methods

Detailed information about key resources is available in S1 Table.

### Animals

C57BL/6J male mice were purchased from Białystok University, Poland. Thy1-GFP(M) (The Jackson Laboratory, JAX:007788, RRID:IMSR_JAX:007788) mutant mice were bred as heterozygotes at Nencki Institute, and PCR genotyped as previously described [42]. αCaMKII-T286A mutant mice were bred as heterozygotes at Nencki Institute, and PCR genotyped as previously described [40]. All mice in the experiments were 10-week old at the beginning of the experiments. The mice were housed in groups of 2 to 6 and maintained on a 12-hour light/dark cycle with food and water ad libitum. All experiments with transgenic mice used approximately equal numbers of males and females. The experiments were undertaken according to the Animal Protection Act of Poland and approved by the I Local Ethics Committee (261/2012 and 829/2019 Warsaw, Poland).

## Contextual fear conditioning (CFC)

The animals were trained in a conditioning chamber (Med Associates, St Albans, USA) in a soundproof box. The chamber floor had a stainless steel grid for shock delivery. Before training, the chamber was cleaned with 70% ethanol, and a paper towel soaked in ethanol was placed under the grid floor. To camouflage background noise in the behavioural room, a white noise generator was placed inside the soundproof box.

On the conditioning day, the mice were brought from the housing room into a holding room to acclimatise for 30 minutes before training. Next, mice were placed in the training chamber, and after a 148-second introductory period, a foot shock (2 seconds, 0.7 mA) was presented. The shock was repeated 5 times, at 90-second intertrial intervals. Thirty seconds after the last shock, the mouse was returned to its home cage. Contextual fear memory was tested and extinguished 24 hours after training by reexposing mice to the conditioning chamber for 30 minutes without US presentation, followed by the second 5-minute test session on the following day. During extensive contextual fear extinction, 30-minute fear extinction sessions were repeated on days 2, 3, 14, 15, and 16. Moreover, mice activity and freezing were tested in context B (Ctx B) on day 17. A video camera was fixed inside the door of the sound attenuating box for the behaviour to be recorded and scored. Freezing behaviour (defined as complete lack of movement, except respiration) and locomotor activity of mice were automatically scored. The experimenters were blind to the experimental groups.

## Stereotactic surgery

Mice were fixed in a stereotactic frame (51503, Stoelting, Wood Dale, IL, USA) and kept under isoflurane anaesthesia (5% for induction, 1.5% to 2.0% during surgery). Adeno-associated viruses, serotype 1 and 2 (AAV1/2), solutions were injected into the dorsal CA1 area at coordinates in relation to Bregma (AP, −2.1 mm; ML, ±1.1 mm; DV, −1.3 mm). Around 450 nl of AAV solutions were injected into the CA1 through a bevelled 26 gauge metal needle, and 10 μl microsyringe (SGE010RNS, WPI, USA) connected to a pump (UMP3, WPI, Sarasota, USA), and its controller (Micro4, WPI, Sarasota, USA) at a rate 50 nl/min. The needle was then left in place for 5 minutes, retracted +100 nm DV, and left for an additional 5 minutes to prevent unwanted spread of the AAV solution. Titers of AAV1/2 were as follows: αCaMKII_PSD-95 (WT):mCherry (PSD-95(WT)): $1.35 \times 10^9$/μl, αCaMKII_PSD-95(S73A):mCherry (PSD-95 (S73A)): $9.12 \times 10^9$/μl), αCaMKII_mCherry (mCherry): viral titer $7.5 \times 10^7$/μl (obtained from Karl Deisseroth's Lab). Mice were allowed to recover from anaesthesia for 2 to 3 hours on a heating pad and then transferred to individual cages where they stayed until complete skin healing, and next, they were returned to the home cages. The viruses were prepared at the Nencki Institute core facility, Laboratory of Animal Models. After training, the animals were perfused with 4% PFA in PBS, and brain sections from the dorsal hippocampus were immunostained for PSD-95 and imaged with Zeiss Spinning Disc confocal microscope (magnification: 10×) to assess the extent of the viral expression and PSD-95 expression.

## Immunostaining

Mice were anaesthetised and perfused with cold phosphate buffer (pH 7.4), followed by 0.5% 4% PFA in phosphate buffer. Brains were removed and postfixed o/n in 4˚C. Brains were kept in 30% sucrose in PBS for 72 hours. Coronal brain sections were prepared using cryosectioning (40 μm thick, Cryostat CM1950, Leica Biosystems Nussloch GmbH, Wetzlar, Germany) and stored in a cryoprotecting solution in −20˚C (PBS, 15% sucrose (Sigma-Aldrich), 30% ethylene glycol (Sigma-Aldrich), and 0.05% NaN₃ (Sigma-Aldrich). Before staining, sections were washed 3 × PBS and blocked for 1 hour at room temperature (RT) in 5% NDS with 0.3%

Triton X-100 in PBS and then incubated o/n, 4˚C with PSD-95 primary antibodies (1:500, Millipore, MAB1598, RRID:AB_11212185) and/or rabbit anti-mCherry primary antibodies (1:500, Abcam, ab167453, RRID:AB_2571870) and/or rabbit P-Ser73_PSD-95 primary antibodies (1:12, Davids Biotechnology, A061). On the second day, slices were washed 3 × PBS with 0.3% Trition X-100 and incubated for 90 minutes with secondary antibodies conjugated with anti-mouse Alexa Fluor 555 (1:500, Invitrogen, A31570, RRID:AB_2536180) and/or anti-rabbit Alexa Fluor 555 (1:500, Invitrogen, A31572, RRID:AB_162543) and/or anti-rabbit Alexa Fluor 647 (1:500, Invitrogen, A31573, RRID:AB_2536183). Slices were then mounted on microscope slides (Thermo Fisher Scientific) and covered with coverslips in Fluoromount-G medium with DAPI (00-4959-52, Invitrogen).

## Phospho-PSD-95(S73)-specific antibody

Phospho-epitope–specific serum against phosphorylated PSD-95(S73) was raised in a rabbit using the synthetic phosphopeptide LERGN(Sp)GLGFS. The antibody was prepared and affinity-purified by Davids Biotechnologie (Regensburg, Germany).

## Confocal microscopy and image quantification

The microphotographs of dendritic spines in the Thy1-GFP(M) mice, fluorescent PSD-95, and phospho-PSD-95(S73) immunostaining were taken on a Spinning Disc confocal microscope (63 × oil objective, NA 1.4, pixel size 0.13 μm × 0.13 μm) (Zeiss, Göttingen, Germany). We took microphotographs (16 bit, z-stacks of 12 to 48 scans; 260 nm z-steps) of 6 dendrites per region per animal from stOri, stRad, and stLM (in the middle of the strata) of dCA1 pyramidal neurons (AP, Bregma from −1.7 to 2.06). The PSD-95 fluorescent immunostaining after AAV overexpression was analysed with Zeiss LSM 800 microscope equipped with Airy-Scan detection (63× oil objective and NA 1.4, pixel size 0.13 μm × 0.13 μm, 8 bit) (Zeiss, Göttingen, Germany). A series of 18 continuous optical sections (67.72 μm × 67.72 μm), at 0.26 μm intervals, were scanned along the z-axis of the tissue section. From every sixth section through dCA1, 6 to 8 z-stacks of microphotographs were taken per animal per region. Each dendritic spine was manually outlined, and the spine area was measured with ImageJ 1.52n software measure tool. Custom-written Python scripts were used to analyse total PSD-95 (mean gray value of the microphotographs: all experiments); PSD-95+ puncta density (per μm of a dendrite) and mean grey value per dendritic spine in Thy1-GFP(M) (Fig 1).

## Serial block-face scanning electron microscopy (SBEM)

Mice were transcardially perfused with cold phosphate buffer (pH 7.4), followed by 0.5% EM-grade glutaraldehyde (G5882 Sigma-Aldrich) with 2% PFA in phosphate buffer (pH 7.4) and postfixed overnight in the same solution. Brains were then taken out of the fixative and cut on a vibratome (Leica VT 1200) into 100-μm slices. Slices were kept in phosphate buffer (pH 7.4), with 0.1% sodium azide in 4˚C. For AAV-injected animals, the fluorescence of exogenous proteins was confirmed in all slices by fluorescent imaging. Then, slices were washed 3 times in cold phosphate buffer and postfixed with a solution of 2% osmium tetroxide (#75632 Sigma-Aldrich) and 1.5% potassium ferrocyanide (P3289 Sigma-Aldrich) in 0.1 M phosphate buffer (pH 7.4) for 60 minutes on ice. Next, samples were rinsed 5 × 3 minutes with double distilled water (ddH$_2$O) and subsequently exposed to 1% aqueous thiocarbohydrazide (TCH) (#88535 Sigma) solution for 20 minutes. Samples were then washed 5 × 3 minutes with ddH$_2$O and stained with osmium tetroxide (1% osmium tetroxide in ddH$_2$O, without ferrocyanide) for 30 minutes in RT. Afterward, slices were rinsed 5 × 3 minutes with ddH$_2$O and incubated in 1% aqueous solution of uranium acetate overnight in 4˚C. The next day, slices were rinsed 5 × 3

minutes with ddH$_2$O, incubated with lead aspartate solution (prepared by dissolving lead nitrate in L-aspartic acid as previously described [85]) for 30 minutes in 60˚C and then washed 5 × 3 minutes with ddH$_2$O, and dehydration was performed using graded dilutions of ice-cold ethanol (30%, 50%, 70%, 80%, 90%, and 2 × 100% ethanol, 5 minutes each). Then, slices were infiltrated with Durcupan resin. A(17 g), B(17 g), and D(0,51 g) components of Durcupan (#44610 Sigma-Aldrich) were first mixed on a magnetic stirrer for 30 minutes, and then 8 drops of DMP-30 accelerator (#45348 Sigma) were added. Part of the resin was then mixed 1:1 (v/v) with 100% ethanol, and slices were incubated in this 50% resin on a clock-like stirrer for 30 minutes in RT. The resin was then replaced with 100% Durcupan for 1 hour in RT, and then 100% Durcupan infiltration was performed o/n with constant slow mixing. The next day, samples were infiltrated with freshly prepared resin (as described above) for another 2 hours in RT and then embedded between flat Aclar sheets (Ted Pella #10501–10). Samples were put in a laboratory oven for at least 48 hours at 65˚C for the resin to polymerise. After the resin hardened, the Aclar layers were separated from the resin-embedded samples, and the dCA1 region was cut out with a razorblade. Caution was taken for the piece to contain minimal resin. Squares of approximately 1 × 1 × 1 mm were attached to aluminium pins (Gatan metal rivets, Oxford Instruments) with very little amount of cyanacrylamide glue. After the glue dried, samples were mounted to the ultramicrotome to cut 1-μm thick slices. Slices were transferred on a microscope slide, briefly stained with 1% toluidine blue in 5% borate and observed under a light microscope to confirm the region of interest (ROI). Next, samples were grounded with silver paint (Ted Pella, 16062–15) and pinned for drying for 4 to 12 hours, before the specimens were mounted into the 3View2 chamber.

## SBEM imaging and 3D reconstructions

Samples were imaged with Zeiss SigmaVP (Zeiss, Oberkochen, Germany) scanning electron microscope equipped with 3View2 chamber using a backscatter electron detector. Scans were taken in the middle portion of dCA1 stOri. From each sample, 200 sections were collected (thickness 60 nm). Imaging settings: high vacuum with EHT 2.9 to 3.8 kV, aperture: 20 μm, pixel dwell time: 3 μs, pixel size: 5 to 6.2 nm. Scans were aligned using the ImageJ software (ImageJ -> Plugins -> Registration -> StackReg) and saved as.tiff image sequence. Next, aligned scans were imported to Reconstruct software, available at http://synapses.clm.utexas.edu/tools/reconstruct/reconstruct.stm (Synapse Web Reconstruct, RRID:SCR_002716). Dendritic spine density was analysed from 3 bricks per animal with the unbiased brick method [44] per tissue volume. Brick dimensions 3 × 3 × 3 μm were chosen to exceed the length of the largest profiles in the data sets at least twice. To calculate the density of dendritic spines, the total volume of large tissue discontinuities was subtracted from the volume of the brick. The density of dendritic spines was normalised to AAV1/2 penetrance.

A structure was considered to be a dendritic spine when it was a definite protrusion from the dendrite, with electron-dense material (representing postsynaptic part of the synapse, PSD) on the part of the membrane that opposed an axonal bouton with at least 3 vesicles within a 50-nm distance from the cellular membrane facing the spine. For 3D reconstructions, PSDs and dendritic spines in one brick were reconstructed for each sample. PSDs were first reconstructed, and, second, their dendritic spines were outlined. To separate dendritic spine necks from the dendrites, a cutoff plane was used approximating where the dendritic surface would be without the dendritic spine. PSD volume was measured by outlining dark, electron-dense areas on each PSD-containing section [45]. The PSD area was measured manually according to the Reconstruct manual. All nonsynaptic protrusions were omitted in this analysis. For multisynaptic spines, the PSD areas and volumes were summed.

## Correlative light-electron microscopy (CLEM)

CLEM workflow was based on a previously established protocol with some modifications [86]. Mice infused with PSD-95(WT) in the CA1 were perfused as described above. Brains were then removed and postfixed o/n in 4˚C. Approximately 100 μm thick brain slices were cut on a vibratome and embedded in low melting point agarose in phosphate buffer and mounted into imaging chambers. mCherry fluorescence in the stRad was photographed using Zeiss LSM800, z-stacks of 60 images (60 μm thick) at 63× magnification. Next, the slice was transferred under the 2P microscope (Zeiss MP PA Setup), where a Chameleon laser was used to brand mark the ROI (laser length 870 nm, laser power 85%, 250 scans of each line). Then, SBEM staining was performed as described above. The resin-embedded hippocampus was then divided into 4 rectangles, and each was mounted onto metal pins to locate the laser-induced marks. SBEM scanned within the laser-marked frame. The fluorescent image was overlaid onto the SBEM image using dendrites and cell nuclei as landmarks using ImageJ 1.48k software (RRID: SCR_003070).

## Electrophysiology

Mice were deeply anaesthetised with Isoflurane, decapitated, and the brains were rapidly dissected and transferred into ice-cold cutting artificial cerebrospinal fluid (ACSF) consisting of (in mM): 87 NaCl, 2.5 KCl, 1.25 NaH2PO4, 25 NaHCO3, 0.5 CaCl2, 7 MgSO4, 20 D-glucose, 75 sacharose equilibrated with carbogen (5% CO2/95% O2). The brain was cut to 2 hemispheres, and 350-μm thick coronal brain slices were cut in ice-cold cutting ACSF with Leica VT1000S vibratome. Slices were then incubated for 15 minutes in cutting ACSF at 32˚C. Next, the slices were transferred to recording ACSF containing (in mM): 125 NaCl, 2.5 KCl, 1.25 $NaH_2PO_4$, 25 $NaHCO_3$, 2.5 $CaCl_2$, 1.5 $MgSO_4$, 20 D-glucose equilibrated with carbogen and incubated for minimum 1 hour at RT.

Extracellular field potential recordings were recorded in a submerged chamber perfused with recording ACSF in RT. The synaptic potentials were evoked with a Stimulus Isolator (A. M.P.I Isoflex) with a concentric bipolar electrode (FHC, CBARC75) placed in the stOri of CA2 on the experiment. The stimulating pulses were delivered at 0.1 Hz, and the pulse duration was 0.3 ms. Recording electrodes (resistance 1 to 4 MΩ) were pulled from borosilicate glass (WPI, 1B120F-4) with a micropipette puller (Sutter Instruments, P-1000) and filled with recording ACSF. The recording electrodes were placed in stOri of dCA1. Simultaneously, a second recording electrode was placed in the stratum pyramidale to measure population spikes. For each slice, the recordings were done in stOri. Recordings were acquired with Multi-Clamp 700B amplifier (Molecular Devices, California, USA), digitised with Digidata 1550B (Molecular Devices, California, USA) and pClamp 10.7 Clampex 10.0 software (Molecular Devices, California, USA). Input/output curves were obtained by increasing stimulation intensity by 25 μA in the range of 0 to 300 μA. All electrophysiological data were analysed with Axo-Graph 1.7.4 software (Axon Instruments, USA). The amplitude of fEPSP, relative amplitude of population spikes and fibre volley were measured.

## Statistics

Data are presented as mean ± standard error of the mean (SEM) for populations with normal distribution or as median ± interquartile range (IQR) for populations with nonnormal distribution. An animal was used as a biological replication in all experiments except for the dendritic spine size distribution analysis. When the data met the assumptions of parametric statistical tests, results were analysed by one- or repeated measures two-way ANOVA, followed by Tukey's or Fisher's post hoc tests, where applicable. Data were tested for normality by using

the Shapiro–Wilk test of normality and for homogeneity of variances by using the Levene's test. For repeated-measure data with missing observation, a linear mixed model was used to analyse the results, followed by pairwise comparisons with Sidak adjustment for multiple comparisons. Areas of dendritic spines and PSDs did not follow normal distributions and were analysed with the Kruskal–Wallis test. Frequency distributions of PSD area to the spine volume ratio were compared with the Kolmogorov–Smirnov test. Correlations were analysed using Spearman correlation (Spearman r ($s_r$) is shown), and the difference between slopes or elevation between linear regression lines was calculated with ANCOVA. Differences between the experimental groups were considered statistically significant if $P < 0.05$. Analyses were performed using the Graphpad Prism 9. Mice were excluded from the analysis only if they did not express the tested virus in the target region.

## Supporting information

**S1 Fig. Synaptic plasticity induced by exposure to neutral context.** Dendritic spines were analysed in 3 domains of dendritic tree of dCA1 area in Thy1-GFP(M) male mice: stOri, stRad, and stLM. **(A)** Experimental timeline and freezing levels of mice from 3 experimental groups: 5US (mice killed 1 day after CFC; $n = 6$), Ctx (mice killed immediately after the second exposure to novel context, no foot shocks were delivered, $n = 5$) and Ext 60' (mice killed 60 minutes after contextual fear extinction session, $n = 6$). **(B)** Representative confocal images of dendrites (GFP) (maximum projections of z-stacks composed of 20 scans) are shown for 3 domains of the dendritic tree. **(C)** Summary of data showing dendritic spine density (repeated-measures ANOVA, effect of training: $F(2, 14) = 1.620$, $P = 0.233$). **(D)** Summary of data showing average dendritic spine area (repeated-measures ANOVA, effect of training: $F(2, 14) = 3.162$, $P = 0.074$). For C, D, each dot represents 1 mouse. **(E)** Representative confocal images of PSD-95 immunostaining (maximum projections of z-stacks composed of 20 scans) are shown for 3 domains of the dendritic tree. **(F)** Summary of data showing total PSD-95 levels (repeated-measures ANOVA with post hoc Tukey test, effect of training: $F(2, 14) = 2.72$, $P = 0.100$; effect of region: $F(1.34, 18.7) = 25.1$, $P < 0.001$; training × region interaction: $F(4, 28) = 2.79$, $P = 0.045$). For C, D, F means ± SEM are shown. The data underlying this figure are available from OSF (https://osf.io/cgfa9/). CFC, contextual fear conditioning; dCA1, dorsal CA1; PSD-95, postsynaptic density protein 95; stLM, stratum lacunosum-moleculare; stOri, stratum oriens; stRad, stratum radiatum.
(TIF)

**S2 Fig. Validation of the viral vectors encoding PSD-95(WT) and PSD-95(S73A). (A)** Experimental timeline. C57BL/6J male mice were stereotactically injected in the dCA1 with AAV1/2 encoding mCherry (Control, $n = 11$) PSD-95(WT) (WT, $n = 11$) or PSD-95(S73A) (S73A, $n = 11$). Twenty-one days later, they were killed. **(B)** Representative confocal scans of the PSD-95 immunostaining in dCA1 strata and **(C)** summary of data showing PSD-95 levels (two-way ANOVA with Tukey's post hoc test, effect of virus: $F(2, 30) = 13.1$, $P < 0.001$). **(D)** Exemplary reconstructions of dendritic spines and their PSDs from SBEM scans in stOri. The grey background rectangles are x = 3 × y = 3 μm. Dendritic spines and PSDs were reconstructed and analysed in tissue bricks (3 × 3 × 3 μm). **(E-G)** Summary of SBEM data showing: **(E)** mean density of dendritic spines (one-way ANOVA with post hoc Tukey test, effect of virus: $F(2, 6) = 34.6$, $P < 0.001$); **(F)** median PSD surface area (Kruskal–Wallis test with Dunn's multiple comparisons test, Kruskal–Wallis statistic = 109, $P < 0.001$), and **(G)** total PSD area per tissue brick (one-way ANOVA, effect of virus: $F(2, 6) = 0.0135$, $P = 0.9870$). The data underlying this figure are available from OSF (https://osf.io/cgfa9/). dCA1, dorsal CA1; PSD, postsynaptic density; PSD-95, postsynaptic density protein 95; S73, Serine 73; SBEM,

serial block-face scanning electron microscopy; stOri, stratum oriens; WT, wild-type.
(TIF)

**S3 Fig. Synaptic plasticity induced in dCA1 during contextual fear extinction training is compensatory. (A)** Experimental timeline. C57BL/6J male mice were stereotactically injected in the dCA1 with AAV1/2 encoding PSD-95(WT) (WT, $n$ = 11) or PSD-95(S73A) (S73A, $n$ = 11). Twenty-one days later, they were trained and killed 24 hours after CFC or immediately after the Extinction session. **(B)** Microphotographs of the brain sections with AAVs expression in dCA1. **(C, D)** Representative fEPSPs evoked by stimuli of different intensities and summary of data in WT and S73A mice after and before fear extinction. **(C)** Input–output functions for stimulus intensity in WT mice (repeated-measures ANOVA, effect of virus × stimulus interaction, $F(12, 456) = 2.73$, $P = 0.001$) and fibre volley recorded in response to increasing intensities of stimulation (repeated-measures ANOVA, effect of virus × stimulus interaction, $F(12, 384) = 0.467$, $P = 0.933$). **(D)** Input–output functions for stimulus intensity in S73A mice (repeated-measures ANOVA, effect of virus × stimulus interaction, $F(12, 456) = 1.50$, $P = 0.120$) and fibre volley recorded in response to increasing intensities of stimulation (repeated-measures ANOVA, effect of virus × stimulus interaction, $F(12, 441) = 0.412$, $P = 0.959$). The numbers of the analysed sections/mice per experimental group are indicated in the legends. Means ± SEM are shown on the graphs. The data underlying this figure are available from OSF (https://osf.io/cgfa9/). CFC, contextual fear conditioning; dCA1, dorsal CA1; fEPSP, field excitatory postsynaptic potential; PSD-95, postsynaptic density protein 95; S73, Serine 73; WT, wild-type.
(TIF)

**S1 Raw Image. The original picture for Fig 3A of the western blot stained with phospho-PSD-95(S73)-specific antibody detects in the hippocampus homogenates proteins with approximately 95 kDA molecular weight.** M, molecular weight marker; N, naive mouse; 5US, mouse that underwent CFC and was killed 24 hours later, Ext15', mouse that underwent CFC and was killed after 15 minutes of a fear extinction session; x, sample not related to the study.
(PDF)

**S1 Table. Key resources.** The table includes information about key materials used in the study.
(DOCX)

## Acknowledgments

The project was carried out using CePT infrastructure financed by the European Union—The European Regional Development Fund within the Operational Program "Innovative economy" for 2007–2013.

## Author Contributions

**Conceptualization:** Kasia Radwanska.

**Data curation:** Magdalena Ziółkowska, Malgorzata Borczyk, Anna Cały, Maria Nalberczak-Skóra, Kacper Łukasiewicz, Tytus Bernaś, Kasia Radwanska.

**Formal analysis:** Magdalena Ziółkowska, Malgorzata Borczyk, Anna Cały, Kamil F. Tomaszewski, Agata Nowacka, Maria Nalberczak-Skóra, Małgorzata Alicja Śliwińska, Kacper Łukasiewicz, Edyta Skonieczna, Tytus Bernaś, Ahmad Salamian.

**Funding acquisition:** Magdalena Ziółkowska, Kasia Radwanska.

**Project administration:** Kasia Radwanska.

**Resources:** Małgorzata Alicja Śliwińska, Tomasz Wójtowicz, Jakub Wlodarczyk.

**Supervision:** Kasia Radwanska.

**Visualization:** Kasia Radwanska.

**Writing – original draft:** Magdalena Ziółkowska, Malgorzata Borczyk, Kasia Radwanska.

**Writing – review & editing:** Magdalena Ziółkowska, Malgorzata Borczyk, Anna Cały, Kamil F. Tomaszewski, Agata Nowacka, Maria Nalberczak-Skóra, Małgorzata Alicja Śliwińska, Kacper Łukasiewicz, Edyta Skonieczna, Tomasz Wójtowicz, Jakub Wlodarczyk, Tytus Bernaś, Ahmad Salamian, Kasia Radwanska.

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
