## [Editor Report · Decision Letter 0]

20 Jan 2023

Dear Dr Radwańska, 

Thank you for submitting your manuscript entitled "Phosphorylation of PSD-95 at Serine 73 in dCA1 is required for extinction of contextual fear" for consideration as a Research Article by PLOS Biology.

Your manuscript, the Review Commons reviews and your point-by-point response has now been evaluated by the PLOS Biology editorial staff, as well as by an academic editor with relevant expertise. I am writing to let you know that we would like to send your revised submission out to the Review Commons reviewers for peer review at PLOS Biology.

Once your full submission is complete, your paper will undergo a series of checks in preparation for peer review. After your manuscript has passed the checks it will be sent out for review. To provide the metadata for your submission, please Login to Editorial Manager (https://www.editorialmanager.com/pbiology) within two working days, i.e. by Jan 22 2023 11:59PM.

Kind regards,

Kris

Kris Dickson, Ph.D., (she/her)

Neurosciences Senior Editor/Section Manager

PLOS Biology

kdickson@plos.org

---

## [Decision Letter · Decision Letter 1]

16 Feb 2023

Dear Dr Radwańska,

Thank you for your patience while your revision plan to your Review Commons submission "Phosphorylation of PSD-95 at Serine 73 in dCA1 is required for extinction of contextual fear" was reviewed by the PLOS Biology editors, and Academic Editor and one of the original Review Commons reviewers (Reviewer 2). We would like to invite you to revise the work to thoroughly address the reviewers' reports as you have outlined. Please note that Reviewer 2 also requests some additional electrophysiology work that we would also like to see you incorporate.

Given the extent of revision needed, we cannot make a decision about publication until we have seen the revised manuscript and your response to the reviewers' comments. Your revised manuscript will be sent for further evaluation by the original Review Commons reviewers when it is resubmitted.

At this stage, your manuscript is under active consideration at our journal; please notify us by email if you do not intend to submit a revision so that we may withdraw it.

**IMPORTANT - SUBMITTING YOUR REVISION**

Your revisions should address the specific points made by each reviewer during the Review Commons review process, along with the additional comments raised by Reviewer 2 below. Please submit the following files along with your revised manuscript:

1. A 'Response to Reviewers' file - this should present a point-by-point response to all of the reviewers' comments (both original Review Commons and ones below), and indicate the changes made to the manuscript. 

*NOTE: In your point-by-point response to the reviewers, please provide the full context of each review from Review Commons and from below. Do not selectively quote paragraphs or sentences to reply to. The entire set of reviewer comments should be present in full and each specific point should be responded to individually, point by point.

*Re-submission Checklist*

*Published Peer Review*

*PLOS Data Policy*

*Blot and Gel Data Policy*

Sincerely,

Kris

Kris Dickson, Ph.D., (she/her)

Neurosciences Senior Editor/Section Manager

PLOS Biology

kdickson@plos.org

REVIEWS:

Reviewer's Responses to Questions

Do you want your identity to be public for this peer review?

Reviewer #1: Yes: Kim Dore (Originally Reviewer 2 at Review Commons)

Reviewer #1: I reviewed this manuscript for Review Commons (I'm reviewer #2). 

The authors responded to my comments in the response letter by saying that they will do all the requested changes and one other experiment I requested (regarding Fig. 6, third comment). If the authors are indeed making all the requested changes, I think this publication would be of interest to PLOS biology readers. 

One thing I'm unsure about is the answer to the first 'major point' I noted. I'm still very concerned about the effects of WT-PSD-95 expression on learning and memory as it's been shown in vitro that it blocks LTP and enhance LTD. 

Even if overpexpression leads to 'only' a 40% increase in PSD-95 in vivo, I would expect it to have an impact on synaptic plasticity. A great way to test this would be to do electrophysiology on acute slices made from control or injected animals. 

If LTP can be induced in PSD-95 injected animals, that would support the results presented and confirm that memory formation is intact in these animals.

I look forward to see how the authors will address all the comments in the revised manuscript.

---

## [Editor Report · Decision Letter 2]

28 Mar 2023

Dear Dr Radwańska,

Thank you for your patience while we considered your revised manuscript "Phosphorylation of PSD-95 at Serine 73 in dCA1 is required for extinction of contextual fear" for publication as a Research Article at PLOS Biology. This revised version of your manuscript has been evaluated by the PLOS Biology editors and the Academic Editor, who feels the manuscript has properly addressed the reviewer comments. 

Therefore, based on our Academic Editor's assessment of your revision, we are likely to accept this manuscript for publication. However, before we can editorially accept your study, we need you to address the following data and other policy-related requests in a revised manuscript that we think should not take very long. 

**EDITORIAL REQUESTS: 

1) TITLE: We think that 'Serine' should not be capitalized in the title. If you agree, we suggest you change the title to: "Phosphorylation of PSD-95 at serine 73 in dCA1 is required for extinction of contextual fear"

2) FINANCIAL DISCLOSURES: Please update your financial disclosures statement to describe the role of any sponsors or funders in the study design, data collection and analysis, decision to publish, or preparation of the manuscript. If the funders had no role in any of the above, include this sentence at the end of your statement: "The funders had no role in study design, data collection and analysis, decision to publish, or preparation of the manuscript

3) DATA REQUEST: Thank you for including the raw western blot images related to your manuscript. Can you please annotate this data, to clearly indicate which figure panel the image corresponds to, to note the loading order, antibody used, and molecular weights? Please add these annotations without obscuring the lanes. Please carefully read our guidelines for how to prepare and upload this data: https://journals.plos.org/plosbiology/s/figures#loc-blot-and-gel-reporting-requirements

4) DATA REQUEST: Thank you also for uploading the raw data underlying your figures on OSF. Can you please reference this dataset in each figure legend? For example, to each figure legend (including supplemental figure legends), you can add the sentence "the data underlying this figure is available from OSF (https://osf.io/cgfa9/)

5) ABSTRACT: Please note that per journal policy, the model system/species studied should be clearly stated in the abstract of your manuscript. Please update it accordingly. 

We expect to receive your revised manuscript within two weeks. 

*Published Peer Review History*

*Press*

Sincerely,

Luke

Lucas Smith, Ph.D.

Associate Editor,

lsmith@plos.org,

PLOS Biology

---

## [Editor Report · Decision Letter 3]

4 Apr 2023

Dear Dr Radwańska,

Thank you for the submission of your revised Research Article "Phosphorylation of PSD-95 at serine 73 in dCA1 is required for extinction of contextual fear" for publication in PLOS Biology. On behalf of my colleagues and the Academic Editor, Eunjoon Kim, I am pleased to say that we can in principle accept your manuscript for publication, provided you address any remaining formatting and reporting issues. These will be detailed in an email you should receive within 2-3 business days from our colleagues in the journal operations team; no action is required from you until then. Please note that we will not be able to formally accept your manuscript and schedule it for publication until you have completed any requested changes.

PRESS

Sincerely, 

Lucas Smith, Ph.D.

Associate Editor

PLOS Biology

lsmith@plos.org